# Heart rate variability and psychosocial symptoms in adolescents and young adults with cancer

**Mallory R. Taylor**[1,2,3]*, **Michelle M. Garrison**[4], **Abby R. Rosenberg**[1,2,3]

**1** Department of Pediatrics, Division of Hematology/Oncology, University of Washington School of Medicine, Seattle, Washington, United States of America, **2** Palliative Care and Resilience Lab, Center for Clinical and Translational Research, Seattle Children's Research Institute, Seattle, Washington, United States of America, **3** Cambia Palliative Care Center of Excellence, University of Washington, Seattle, Washington, United States of America, **4** Department of Health Services, University of Washington, Seattle, Washington, United States of America

* molly.taylor@seattlechildrens.org

**Data Availability Statement:** All relevant data are within the manuscript and its Supporting information files.

**Funding:** MT was supported by a T32 training grant from the NIH/NHLBI (T32HL125195-05) as

## Abstract

### Background

Heart Rate Variability (HRV) is a valid, scalable biomarker of stress. We aimed to examine associations between HRV and psychosocial outcomes in adolescents and young adults (AYAs) with cancer.

### Methods

This was a secondary analysis of baseline data from a randomized trial testing a resilience intervention in AYAs with cancer. Two widely used HRV metrics, the standard deviation of normal to normal beats (SDNN) and root mean square of successive differences (RMSSD), were derived from electrocardiograms. Patient-reported outcome (PRO) survey measures included quality of life, anxiety, depression, distress, and resilience. Linear regression models were used to test associations between HRV and PRO scores. The Wilcoxon rank sum test was used to test differences in median HRV values among participant subgroups.

### Results

Among the n = 76 patients with available electrocardiograms, the mean age was 16 years (SD 3y), 63% were white, and leukemia/lymphoma was the most common diagnosis. Compared to healthy adolescents, AYAs with cancer had lower median HRV (SDNN [Females: 31.9 (12.8–50.7) vs 66.4 (46.0–86.8), p<0.01; Males: 29.9 (11.5–47.9) vs 63.2 (48.4–84.6), p<0.01]; RMSSD [Females: 28.2 (11.1–45.5) vs 69.0 (49.1–99.6), p<0.01; Males: 27.9 (8.6–48.6) vs 58.7 (44.8–88.2), p<0.01]). There was no statistically significant association between PRO measures and SDNN or RMSSD in either an unadjusted or adjusted linear regression models.

well as a Conquer Cancer Young Investigator Award, supported by Women Leaders in Oncology. Any opinions, findings, and conclusions expressed in this material are those of the author(s) and do not necessarily reflect those of the American Society of Clinical Oncology®, Conquer Cancer®, or Women Leaders in Oncology. The funders had no role in study design, data collection and analysis, decision to publish, or preparation of the manuscript.

**Competing interests:** The authors have declared that no competing interests exist.

## Conclusion

In this secondary analysis, we did not find an association between HRV and psychosocial PROs among AYAs with cancer. HRV measures were lower than for healthy adolescents. Larger prospective studies in AYA biopsychosocial research are needed.

## Introduction

Psychological distress is prevalent during adolescent development, and adolescents and young adults (AYAs) with serious illness like cancer experience even higher rates of depression, post-traumatic stress, and suicide [1–3]. Poor mental health directly translates to adverse cancer-related outcomes, including increased physical symptom burden [4] and poorer survival [5, 6]. In contrast, positive psychological well-being has been associated with protective effects, including lower risk of relapse and cancer-related mortality [7]. Identifying the mechanisms underlying these biobehavioral connections is a growing focus of investigation and may offer a novel tool for symptom-based risk stratification and intervention.

Psychosocial and environmental stressors can activate a cascade of signaling pathways that have overlapping behavioral and biologic implications. For example, a perceived threat (such as a new cancer diagnosis) triggers the hypothalamic-pituitary-adrenal (HPA) axis to ultimately secrete glucocorticoids into circulation, which have well-documented metabolic and immunomodulatory effects that could influence cancer-related outcomes [8]. The autonomic nervous system (ANS) is another key physiologic mediator of the relationship between patient experience and outcomes in biobehavioral oncology. Activation of the parasympathetic and sympathetic branches of the ANS has been associated with symptoms of anxiety, depression, and fatigue in patients with cancer [9, 10]. Conversely, interruption of sympathetic adrenergic signaling using β-blocker medications is associated with improved metastatic and inflammatory biomarkers, as well as lower cancer-specific mortality in patients with breast cancer [11–13].

Heart rate variability (HRV), which is the physiologic fluctuation between successive heartbeats, is a surrogate marker of ANS status and has been widely applied in both social science and biomedical settings [14, 15]. HRV is measured by capturing electrical signals from the heart using electrocardiograms (ECGs), Holter monitors, pulse oximeters, or newer wearable devices and smartwatches. There are published guidelines for measuring and interpreting HRV [16], and although gold standard for HRV measurement is a 24-hour recording, components of HRV can reliably be determined from short (1 or 5 minute) or ultra-short (10 second) monitoring [17, 18].

Lower HRV, signaling reduced autonomic flexibility, is associated with important outcomes including infection, mortality [19], depression and anxiety disorders [15, 20], and psychosocial stress [21]. Adult patients with cancer are known to have lower HRV compared to healthy controls, with this difference more pronounced in those with advanced cancer [22]. Reduced HRV has also been reported in young children with leukemia [23, 24] and dysautonomia indexed by HRV has been documented in adult survivors of childhood cancer [25]. Importantly, the degree of HRV reduction may predict cancer-related fatigue and chronic pain [26, 27] as well as cancer progression and overall survival [28]. The precise role of the ANS in these conditions is incompletely understood, but the direct interplay between psychosocial risk factors, the ANS, and inflammation provide a biologically plausible explanation for the connection between mental and physical symptoms in cancer [29].

To date, nearly all oncology research using HRV has focused on adults, aside from a small number of studies in younger patients with leukemia [23]. There have been no explicit studies investigating the utility of HRV as a potential stress biomarker in older AYA oncology patients. Given the extreme physical and psychosocial stress associated with cancer diagnosis and treatment in adolescents and young adults [1–3], HRV measurement may be particularly helpful as a simple, non-invasive assessment of patient wellbeing and function. Here, we present a cross-sectional secondary analysis of baseline HRV and patient-centered psychosocial outcomes from a randomized trial among AYAs with cancer.

## Materials and methods

This was a secondary analysis of a completed randomized controlled trial (RCT) testing the Promoting Resilience in Stress Management (PRISM) intervention in AYAs with cancer (NCT02340884). We conducted a cross-sectional analysis of available baseline data prior to receipt of the PRISM intervention [30]. Heart rate variability (SDNN and RMSSD) was our independent variable, and patient-reported psychosocial measures served as our dependent variable. We hypothesized that lower HRV would be associated with adverse psychosocial states (higher anxiety, depression, distress; and lower quality of life and resilience). The Seattle Children's Institutional Review Board approved this study.

### Participants and setting

The Phase II PRISM Randomized Trial was conducted at a single institution (Seattle Children's Hospital) from January 2015 to October 2016. Eligible patients were English speaking, aged 12–25 years, and diagnosed with a new or relapsed/refractory cancer requiring treatment. Demographic and disease-related variables were requested in surveys and collected from the medical record of consented participants. Of the n = 92 AYAs with baseline patient reported psychosocial measures, n = 76 also had available ECGs to derive HRV and were included in the present analysis. Of the n = 16 participants with missing ECGs, 7 did not require ECGs for their specific treatment plan, and there was no documented reason for omission in 9 cases.

### Measures

**Heart rate variability [31].**   HRV was derived using 10 second 100Hz ECGs obtained at the time of cancer diagnosis or relapse as part of routine clinical care. These screening ECGs were obtained as part of patients' medical care and were not part of the clinical trial. Individual paper ECGs were retrospectively extracted from the medical record and scanned into a digital format using a high-resolution scanner. These digital ECG files were then uploaded into an image processing software (WebPlotDigitizer [32]) to extract R-R intervals from the ECG tracing images [33]. R-waves were first identified using the software algorithm and then manually reviewed for accuracy and artifacts. The discrete R-R intervals (in milliseconds) could then be identified from the image and converted to their numerical form. Using the open-source R software, RHRV [34], we then derived the two most widely used time domain parameters: standard deviation of normal to normal beats (SDNN) and root mean square of successive differences (RMSSD) per published guidelines [16, 35]. These two HRV measures were chosen because 1) they are commonly used in behavioral and clinical research; 2) they can be derived from standard ECGs [17] and 3) published age-appropriate normative values exist [36].

**Psychological variables.**   All participants were invited to complete a survey consisting of AYA age–validated instruments upon enrollment and received a $25 gift card upon survey completion.

*Pediatric Quality of Life (PedsQL) Generic and Cancer Module teen reports* [37, 38]. The PedsQL 4.0 Generic and 3.0 Cancer Modules include a combined total of 50 items evaluating QOL of AYAs with cancer. Subscales assess physical, emotional, social, and school well-being, plus cancer-related domains such as pain, nausea, procedural anxiety, and perceived physical appearance. In healthy populations, a score of <70 is considered at risk for poor QOL. In patients with cancer, mean scores for the Generic and Cancer PedsQL Modules are reported at 70.9 (SD 17.2) [37] and 65.3 (SD 16.3) [30], respectively.

*Hospital Anxiety and Depression Scale (HADS)* [39]. The HADS assesses depressive and anxious symptoms in patients with serious illness. It has been validated in AYAs with chronic illness [40] as well as AYA cancer survivors [41]. The mean score for adolescents with cancer is 11 (SD 6.2) [30]. A 'case' of anxiety and depression is defined as $\geq 8$ points.

*Connor-Davidson Resilience Scale (CD-RISC)* [42]. The CD-RISC is a reliable and widely used instrument to measure inherent resiliency. The 10-item instrument has been used in diverse populations including adolescents, parents, and patients with cancer. The mean score among healthy US adults is 31.8 (SD 5.4) [42] and 28 (SD 5.8) in adolescents with cancer [30].

*Kessler-6 general psychological distress scale (K6)* [43]. This 6-item scale measures level of psychological distress experienced in the past month. The instrument has been extensively cross validated, including among adolescents. The average score for healthy adolescents is 5.8 (SD 4.7) [44], and 7.0 (SD 4.7) [30] for adolescents with cancer. Previous studies have shown that scores > 6 are consistent with high distress and those $\geq 13$ meet criteria for serious or debilitating psychological distress [43].

## Data analysis

In this secondary analysis, we used baseline ECGs and survey data to examine the relationship between HRV parameters and patient-reported quality of life and resilience, as well as symptoms of anxiety, depression, and distress. We summarized these baseline measures using means/medians, standard deviations, frequencies, and proportions. All variables were reported as continuous, with some variables converted to ordinal or dichotomous when appropriate using clinically relevant cut points (e.g., a HADS depression subscale score of $\geq 8$). We conducted linear regression modeling to test associations between HRV and PRO scores. In adjusted models, we controlled for age, gender, cancer type (Leukemia/Lymphoma or CNS/Non-CNS Solid Tumors), and cancer status (relapsed or newly diagnosed). Patient age was transformed into an ordinal variable based on age groups thought to be the most similar developmentally and physiologically. Exploratory stratified analyses were also performed to assess the HRV-PRO relationship among differing age categories, gender, cancer type, and cancer relapse status. Additionally, we compared median HRV values of participants to published normative data for sex-matched healthy adolescents [31], as well as among patients with relapsed versus newly diagnosed cancer in our sample using the Wilcoxon rank sum test. All data analysis was conducted using Stata 14 software (StataCorp, College Station, TX).

## Results

There were 76 patients with both ECGs and surveys at baseline. Just under half of participants were female, and the mean age at study entry was 16 years (SD 3 years) (Table 1). The most common cancer diagnosis was leukemia/lymphoma, and most participants identified as white. On average, surveys were collected 7 days after ECGs (rage 85 days before to 64 days after).

**Table 1. Patient characteristics.**

|  | Total N = 76 |
|---|---|
| **Sex** |  |
| Female | 43% |
| **Age Range** | 12-25y |
| Mean (SD) | 16y (3y) |
| 12-15y | 43% |
| 16-19y | 43% |
| 20-25y | 13% |
| **Race** |  |
| White | 63% |
| Black/African American | 3% |
| Asian | 7% |
| Other* | 28% |
| **Cancer Diagnosis** |  |
| Leukemia/Lymphoma | 75% |
| CNS Tumor | 3% |
| Non-CNS Solid Tumor | 22% |
| **Psychological Instrument Score** | **Mean (SD)** |
| HADS Total | 11.1 (6.3) |
| HADS Depression | 5.1 (3.4) |
| Above clinical cutoff on HADS Depression | 29% |
| HADS Anxiety | 6.0 (3.6) |
| Above clinical cutoff on HADS Anxiety | 28% |
| CD-RISC | 28.9 (6.0) |
| Kessler-6 | 6.9 (4.8) |
| Above clinical cutoff | 49% |
| Quality of Life (PedsQL) | 60.3 (19.1) |
| Cancer Specific QOL | 65.6 (16.8) |
| **Elapsed time between survey and ECG**** |  |
| **Mean (range)** in days | 7.7 (-85–64) |

*Other = mixed race, missing, or other.

**Three outliers were removed with gaps of >120 days between survey and ECG.

## Psychological instrument results

At baseline, AYA participants reported a mean HADS score of 11.1 (SD 6.3), with nearly one-third of participants meeting criteria for clinically relevant anxiety or depression. Mean psychological distress score was 6.9 (SD 4.8), with 49% meeting criteria for elevated distress. The mean resilience score was 28.9 (SD 6), and general and cancer specific QOL scores were 60.3 (19.1) and 65.6 (SD 16.8), respectively.

**HRV results.** The median values (IQR) for SDNN and RMSSD were 30.9 (12.7–50.3) and 31.2 (20.4–38.6), respectively (Table 2). Compared to published values for healthy adolescents, participants had statistically significantly lower median SDNN [Females: 31.9 (12.8–50.7) vs 66.4 (46.0–86.8), p<0.01; Males: 29.9 (11.5–47.9) vs 63.2 (48.4–84.6), p<0.01] and RMSSD [Females: 28.2 (11.1–45.5) vs 69.0 (49.1–99.6), p<0.01; Males: 27.9 (8.6–48.6) vs 58.7 (44.8–88.2), p<0.01]. Newly diagnosed patients tended to have higher median HRV values [SDNN = 33.0 (13.2–50.4), RMSSD = 29.5 (11.6–47.6)] than those patients who enrolled in the

**Table 2. Heart rate variability measures.**

| | SDNN median (IQR) | RMSSD median (IQR) |
|---|---|---|
| **All patients (n = 76)** | 30.9 (12.7–50.3) | 31.2 (20.4–38.6) |
| **Age** | | |
| 12–15 (n = 33) | 25.7 (12.8–52.9) | 23.0 (10.6–40.7) |
| 16–19 (n = 33) | 39.5 (8.8–49.0) | 27.9 (8.1–47.7) |
| 20–25 (n = 10) | 34.3(17.4–47.9) | 30.9 (15.8–56.9) |
| **Sex** | | |
| M (n = 43) | 29.9 (11.5–47.9), p <0.01* | 27.9 (8.6–48.6), p <0.01* |
| F (n = 33) | 31.9 (12.8–50.7), p <0.01* | 28.2(11.1–45.5), p <0.01* |
| **Cancer Type** | | |
| Leukemia/Lymphoma (n = 57) | 34.2 (12.8–50.7) | 28.2 (12.2–45.5) |
| Non-CNS Solid Tumor (n = 17) | 28.2 (17.4–46.7) | 30.8 (8.1–48.6) |
| CNS Solid Tumor (n = 2) | 10.1 (7.0–13.2) | 6.7 (5.1–8.1) |
| **Cancer Status** | | |
| Newly diagnosed (n = 58) | 33.0 (13.2–50.4) | 29.5 (11.6–47.6) |
| Relapsed (n = 18) | 23.6 (10.2–50.2), p = 0.47** | 20.8 (10.0–43.1), p = 0.35** |

All results reported in milliseconds.

*Compared to sex-matched published normative values in healthy adolescents.

**Compared to newly diagnosed patients.

trial at the time of relapse [SDNN = 23.6 (10.2–50.2), RMSSD = 20.8 (10.0–43.1)], but this was not statistically significant. Fig 2 illustrates HRV measures stratified by age, gender, cancer type, and cancer relapse status. Additionally, values for both SDNN and RMSSD demonstrated a right skew, with more patients falling on the lower end of the HRV spectrum (Fig 1).

*Associations between HRV and PROs.* There were no linear associations between either measure of HRV and baseline anxiety, depression, distress, quality of life, or resilience. There was no statistically significant association between PRO measures and SDNN or RMSSD in either an unadjusted or adjusted linear regression model (Table 3, Fig 2). In exploratory stratified analyses of patient-reported anxiety and depression scores there were no statistically significant relationships among other subgroups (S1 Table). However, the association between depression and RMSSD approached statistical significance in females (beta coefficient = 0.54, p = 0.09), as did depression and SDNN in patients aged 20–25 years (beta coefficient = 1.06, p = 0.09).

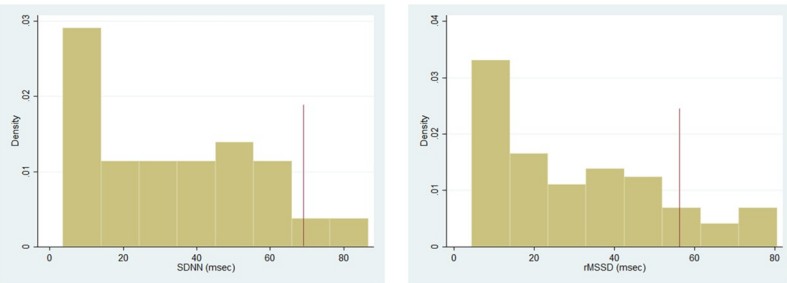

**Fig 1. Histograms showing the distribution of SDNN and RMSSD for the entire cohort of n = 76 participants.** Vertical red lines represent normal median values for healthy adolescents.

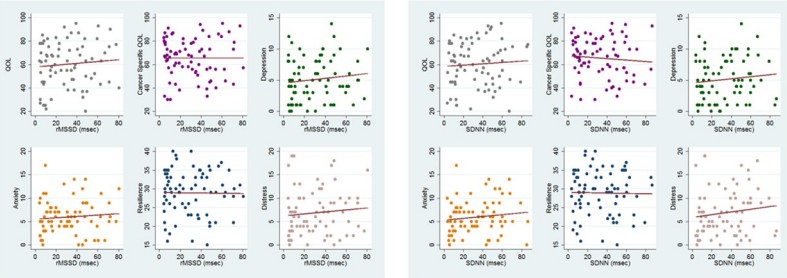

**Fig 2. Scatterplots of the two HRV measures SDNN and RMSSD plotted against patient reported psychosocial symptoms with simple linear regression fitted lines in red.**

## Discussion

In this exploratory post-hoc analysis, we did not find a statistically significant association between heart rate variability and psychosocial patient reported outcomes in AYAs with cancer. However, median HRV parameters were substantially lower than in the healthy adolescent population, which may have broader implications for overall well-being. Results of this study provide baseline normative HRV values in the largest published cohort of AYA oncology patients to date. Given the growing interest in using HRV as a measure of ANS function in supportive care research, documenting the normative values and distribution is an important first step in building a larger program of research in biopsychosocial AYA oncology.

Cancer treatment, especially chemotherapy agents like microtubule inhibitors and anthra-cyclines, can directly damage the nervous and cardiac systems and cause autonomic dysfunc-tion. It is therefore not surprising that reduced HRV has been documented in the leukemia population (where Vincristine and Doxorubicin are often present in treatment regimens) [24] and adult survivors of childhood cancers [25]. Indeed, our data suggest there could be a more

**Table 3. Association of heart rate variability with patient reported outcomes.**

| | SDNN | | | RMSSD | | |
|---|---|---|---|---|---|---|
| *Model 1* | **β coefficient** | **95% CI** | **p-value** | **β coefficient** | **95% CI** | **p-value** |
| *Anxiety* | 0.2 | [-0.2–0.6] | 0.3 | 0.1 | [-0.2–0.5] | 0.46 |
| *Depression* | 0.16 | [-0.2–0.5] | 0.36 | 0.2 | [-0.2–0.6] | 0.32 |
| *Distress* | 0.3 | [-0.2–0.8] | 0.25 | 0.2 | [-0.3–0.7] | 0.42 |
| *Resilience* | -0.05 | [-0.7–0.6] | 0.89 | -0.03 | [-0.7–0.6] | 0.93 |
| *General QOL* | 0.6 | [-1.4–2.6] | 0.57 | 0.8 | [-1.3–2.8] | 0.45 |
| *Cancer-specific QOL* | -0.6 | [-2.4–1.1] | 0.48 | -0.02 | [-1.8–1.8] | 0.99 |
| *Model 2* | | | | | | |
| *Anxiety* | 0.17 | [-0.2–0.5] | 0.35 | 0.2 | [-0.2–0.5] | 0.42 |
| *Depression* | 0.1 | [-0.2–0.5] | 0.44 | 0.2 | [-0.2–0.5] | 0.3 |
| *Distress* | 0.2 | [-0.2–0.7] | 0.3 | 0.2 | [-0.3–0.7] | 0.4 |
| *Resilience* | -0.02 | [-0.6–0.5] | 0.94 | -0.1 | [-0.7–0.5] | 0.71 |
| *General QOL* | 0.7 | [-1.0–2.5] | 0.42 | 0.6 | [-1.2–2.4] | 0.5 |
| *Cancer-specific QOL* | -0.4 | [-1.8–0.9] | 0.51 | -0.2 | [-1.5–1.2] | 0.86 |

Linear regression models with patient reported outcomes as primary outcome of interest, and SDNN or RMSSD as the predictor of interest. Model 1 = unadjusted model, Model 2 = adjusted for Age, Gender, and Cancer Relapse Status.

pronounced HRV reduction in the relapsed patients, likely representing treatment effect in addition to the compounded psychosocial stress of multiple cancer diagnoses.

However, newly diagnosed patients also had median HRV parameters that appeared lower than healthy adolescents, raising the question of other non-treatment-related sources for autonomic dysfunction that may come with a new cancer diagnosis. These could include disruption of normal neuro-cardiovascular physiology by cancer itself (e.g. a large mass in the mediastinum or catecholamine-secreting tumor) or the intensified psychosocial stressor of the diagnosis in the AYA population (and accompanying social, developmental, financial, and existential threats). As HRV is easily measured through non-invasive methods, it could provide an innovative tool to help risk stratify patients who may require additional forms of supportive care.

Participants in this study were enrolled on a larger randomized trial testing the resilience intervention, PRISM. Primary results from this trial indicated the intervention was associated with improved resilience and cancer-specific quality of life, as well as decreased distress [30]. Because we did not have parallel longitudinal screening ECG recordings in our patient cohort, we were unable to evaluate possible effects of the PRISM intervention on HRV. However, understanding the physiologic correlates of psychosocial intervention is an important area for future investigation, particularly in AYA oncology.

There are several possibilities for why we did not find a relationship between HRV and psychological states when this has been reported in other adolescent populations [15]. Importantly, as a secondary analysis of a larger trial, our study was not powered to detect our associations of interest. Additionally, ECGs were not obtained as part of a strict protocol to control known external influences on basal HRV such as time of day, recent physical activity, and controlled respiration [37, 45]. Nonetheless, the use of baseline ECGs (as opposed to ECGs obtained for clinical concern) in a large trial of AYA oncology patients provides a unique opportunity for data collection to inform subsequent study design.

There were some specific HRV-PRO relationships that approached statistical significance and warrant further examination in larger prospective studies. For example, SDNN and depression in female patients, as well as those patients in the older age group (20-25y). Future studies should focus on collecting extended (24h) HRV recordings in parallel with psychosocial patient reported outcome measures. The optimal study protocol would also avoid data collection during acute medical situations that could confound HRV results in patients with cancer (e.g., serious infection, chemotherapy toxicity, significant cardiac medication adjustments).

This study has additional key limitations. We had a relatively small, demographically homogeneous sample, and thus our results may not be generalizable to other populations. Given the nature of multiple comparisons in our analysis, there was also an increased risk for bias in our results. Additionally, there are other variables known to contribute to baseline HRV, such as physical fitness level, that were not collected as part of this study and thus could not be accounted for in the analysis. However, despite these limitations, this is the first study to report baseline HRV data in the adolescent cancer population. This provides a useful framework to develop larger prospective studies in biobehavioral AYA oncology. Future research should include longer protocolized HRV measurement, as well as collect HRV and psychosocial PRO data in parallel to test its utility as a biomarker of well-being in this population.

## Conclusions

In this secondary analysis, we did not find evidence of an association between HRV measures and patient-reported psychosocial outcomes. However, the implications for reduced HRV in AYA oncology patients compared to healthy adolescents warrants further investigation.

Exploring the physiologic underpinnings of patient-centered outcomes could offer novel intervention strategies in patients with serious illness.

## Supporting information

**S1 Table. Association of heart rate variability with depression and anxiety stratified by sex, cancer type and age.** Association of patient reported anxiety and depression with HRV measures stratified by gender, cancer type, and age using linear regression models. Models adjusted for age and cancer type when not the stratum of interest. Coefficients interpreted as change in PRO score for every 10msec change in HRV. CNS and non-CNS solid tumor categories collapsed for analysis.
(DOCX)

**S1 Data.**
(XLSX)

## Acknowledgments

Michael A. Muldoon for technical assistance and general support.

## Author Contributions

**Conceptualization:** Mallory R. Taylor.

**Data curation:** Mallory R. Taylor.

**Formal analysis:** Mallory R. Taylor, Michelle M. Garrison.

**Methodology:** Mallory R. Taylor, Michelle M. Garrison, Abby R. Rosenberg.

**Resources:** Abby R. Rosenberg.

**Supervision:** Michelle M. Garrison, Abby R. Rosenberg.

**Writing – original draft:** Mallory R. Taylor.

**Writing – review & editing:** Michelle M. Garrison, Abby R. Rosenberg.

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
