## [Decision Letter · Decision Letter 0]

7 Jul 2021

PONE-D-21-16445

Heart Rate Variability and Psychosocial Symptoms in Adolescents and Young Adults with Cancer

PLOS ONE

Dear Dr. Taylor,

Thank you for submitting your manuscript to PLOS ONE. After careful consideration, we feel that it has merit but does not fully meet PLOS ONE’s publication criteria as it currently stands. Therefore, we invite you to submit a revised version of the manuscript that addresses the points raised during the review process.

We look forward to receiving your revised manuscript.

Kind regards,

Michael Kaess, M. D.

Academic Editor

PLOS ONE

Journal Requirements:

Reviewers' comments:

Reviewer's Responses to Questions

**Comments to the Author**

1. Is the manuscript technically sound, and do the data support the conclusions?

Reviewer #1: Yes

Reviewer #2: Yes

2. Has the statistical analysis been performed appropriately and rigorously? 

Reviewer #1: No

Reviewer #2: Yes

3. Have the authors made all data underlying the findings in their manuscript fully available?

Reviewer #1: Yes

Reviewer #2: No

4. Is the manuscript presented in an intelligible fashion and written in standard English?

Reviewer #1: Yes

Reviewer #2: Yes

5. Review Comments to the Author

Reviewer #1: Requires revision

This is a very interesting study and certainly worthy of publication. However, the presentation needs improving.

General issues

As indicated above, the presentation needs improving as the tabular presentations do not convey much meaning. For example, the ß coefficients of Table 4 (non-of which are statistically significant) do not give an immediate expression of how important (or not) they are as they refer to the change in PRO for unit change of (say) SDNN. Thus, for Anxiety ß = 0.02 for a 1 ms increase in SDNN whereas a 10ms increase of SDNN would bring a 0.02 � 10 = 0.2 change in PRO. Think of plotting the weights (the y variable) of 10 children whose ages (the x variable) range from 5 to 15 years using days, months or weeks, rather than years. For the first of these analyses the ß coefficient is very small, the second larger, the third larger still but not as large as the fourth. So, looking at Figure 2, a 10ms x-scale might be a better choice for the analysis of this study. This of course would not change the conclusions but ease the interpretation.

More importantly, Table 4 and Figure 2 (interesting though they are) do not give much indication of the relationship between the HRV measure and the PRO. What is needed are some scatter plots of HRV against PRO for all the 76 patients in the study. That is 6 plots for SDNN and 6 for RMSSD. Several statistical packages allow such plots to be collated into two Figures of 6 panels. The fitted simple linear regression model fitted should be superimposed on each panel. Such plots would allow the investigators, and PLoS readers, to get a general impression of what is going on. Are the points close to the fitted lines? Is the relationship linear? Are there any unusual features?

Breaking these down into patient subgroups could then be done using multiple regression techniques with patient characteristics as covariates.

Specific issues

Page Line

2 14-16 These do not give the magnitudes of the differences there are between these patients and their sex-matched population. In any event, I am not sure this analysis needs to be conducted for the purpose of this study.

2 21 As indicated by the authors, the information in this study should form the basis of any proposed ‘Larger prospective study’. However, all statistical tests are reported as non-significant but the lack of significance could be due to (no or a very small effect) or that there are insufficient numbers in this study. So, are any differences observed by the authors in this study suggestive of clinically important effects that could be demonstrated in a larger study. If so, it would be useful to indicate these in their Discussion section.

5 & 6 Table 1 & Line 12 My understanding is that ‘normal or reference ranges’ are usually indicated by mean � 2SD which encompasses most of the data, and not the ICR which only covers the middle 50% of the possible data values.

7 10 Suggest replace ‘7’ by ‘7.0’

Reviewer #2: Title: "Heart Rate Variability and Psychosocial Symptoms in Adolescents and Young Adults with Cancer"

Summary

In their manuscript, the authors describe secondary analyses considering HRV in association with psychosocial outcomes in adolescents and young adults with cancer, as well as a comparison of HRV measures between oncology patients and sex- and age-matched population norms. Analyses were conducted using clinical baseline data from a randomized intervention study and cardiac autonomic measures (calculated in the time-domain) acquired using ultra-short (10 sec) ECG recordings. While the authors reported no evidence for an association between cardiac autonomic measures and any of the psychosocial outcomes, they report significant deviations of SDNN and RMSSD values in newly diagnosed and relapsed cancer patients compared to sex- and age-matched population norms.

This concise article treats an important topic and has the potential to drive further research evaluating autonomic biomarkers of risk stratification in young individuals with cancer.

Yet, I think that currently the article is lacking some in-depth elaboration, while certain figures and tables might be revised / removed.

In the comments below are suggestions to the authors with the aim to help improve the manuscript.

Major comments

1). I would suggest to entirely remove Figure 1 from the manuscript. Instead, the authors should elaborate on the supposed pathways in the according paragraph of the introduction section, while citing relevant literature.

2). I was wondering whether the authors checked for differences in demographics / psychosocial outcomes between patients with and without available ECG recordings?

2). Reporting HRV measurement, the authors may consider following and citing current guidelines: Quintana, D., Alvares, G. & Heathers, J. Guidelines for Reporting Articles on Psychiatry and Heart rate variability (GRAPH): recommendations to advance research communication. Transl Psychiatry 6, e803 (2016). https://doi.org/10.1038/tp.2016.73

3). Table 1 seems redundant, and I would suggest to remove it from the manuscript. In addition, Table 4 could be integrated in the supplementary table. Figure 2 also seems largely redundant, instead, the additional information provided could be integrated in Table 3. Comparisons with population norms might be depicted instead.

4). Can the authors please clarify whether ECG recordings were collected before, at the same time as, or after psychosocial measures, and how much time elapsed between measurements?

6). In their discussion, the authors state "Importantly, as a secondary analysis of a larger trial, our study was not powered to detect our associations of interest". I was wondering whether the authors could explicate how the respective associations of interest should be investigated in future studies?

7). In the discussion the authors also state "However, newly diagnosed patients also had median HRV parameters that appeared lower than healthy adolescents, raising the question of other non-treatment-related sources for autonomic dysregulation." I was wondering whether the authors might enrich their discussion by elaborating on potential "non-treatment-related sources for autonomic dysregulation" in this population, highlighting important research gaps.

Minor comments

8). I was wondering whether the authors checked for differences in demographics / psychosocial outcomes between patients with and without available ECG recordings?

9). The R-package used (RHRV) should be properly cited to acknowledge the authors’ work:

Leandro Rodriguez-Linares, Xose Vila, Maria Jose Lado, Arturo Mendez, Abraham Otero and Constantino Antonio Garcia (2019). RHRV: Heart Rate Variability Analysis of ECG Data. R package version 4.2.5. https://CRAN.R-project.org/package=RHRV

10). For group comparisons, besides p-values the authors may also report according effects sizes.

6. PLOS authors have the option to publish the peer review history of their article (what does this mean?). If published, this will include your full peer review and any attached files.

Reviewer #1: **Yes: **David Machin

Reviewer #2: No

---

## [Author Response · Author response to Decision Letter 0]

4 Aug 2021

We have provided our de-identified data and uploaded to the supplemental files. 

Reviewer 1 Comments:

Comment: As indicated above, the presentation needs improving as the tabular presentations do not convey much meaning. For example, the ß coefficients of Table 4 (non-of which are statistically significant) do not give an immediate expression of how important (or not) they are as they refer to the change in PRO for unit change of (say) SDNN. Thus, for Anxiety ß = 0.02 for a 1 ms increase in SDNN whereas a 10ms increase of SDNN would bring a 0.02 � 10 = 0.2 change in PRO. Think of plotting the weights (the y variable) of 10 children whose ages (the x variable) range from 5 to 15 years using days, months or weeks, rather than years. For the first of these analyses the ß coefficient is very small, the second larger, the third larger still but not as large as the fourth. So, looking at Figure 2, a 10ms x-scale might be a better choice for the analysis of this study. This of course would not change the conclusions but ease the interpretation.

Change: We have adjusted the presentation of the data to make the ß coefficients easier to interpret in Table 3 (formerly Table 4).

Comment: More importantly, Table 4 and Figure 2 (interesting though they are) do not give much indication of the relationship between the HRV measure and the PRO. What is needed are some scatter plots of HRV against PRO for all the 76 patients in the study. That is 6 plots for SDNN and 6 for RMSSD. Several statistical packages allow such plots to be collated into two Figures of 6 panels. The fitted simple linear regression model fitted should be superimposed on each panel. Such plots would allow the investigators, and PLoS readers, to get a general impression of what is going on. Are the points close to the fitted lines? Is the relationship linear? Are there any unusual features?

Change: We have now created the scatterplot panels with the fitted simple linear regression model overlayed to facilitate easier visualization of the data (Figure 2).

Comment: Breaking these down into patient subgroups could then be done using multiple regression techniques with patient characteristics as covariates.

Change: We agree this analysis is informative, and so have included these subgroup multiple regression results in the supplemental information (Supplemental Table 1). Given the relatively small sample size for each patient subgroup, we have limited graphical representations to the full cohort as above.

Comment: Page 2, lines 14-16 These do not give the magnitudes of the differences there are between these patients and their sex-matched population. In any event, I am not sure this analysis needs to be conducted for the purpose of this study.

Change: We have now included the magnitudes of the differences in the abstract (as they are reported in the results), as we feel highlighting the differences in HRV between healthy adolescents and those with cancer is an important adjunct to the other analyses. [Page 2]

Comment: Page 2, Line 21 As indicated by the authors, the information in this study should form the basis of any proposed ‘Larger prospective study’. However, all statistical tests are reported as non-significant but the lack of significance could be due to (no or a very small effect) or that there are insufficient numbers in this study. So, are any differences observed by the authors in this study suggestive of clinically important effects that could be demonstrated in a larger study. If so, it would be useful to indicate these in their Discussion section.

Change: We have included some examples of specific relationships that warrant further study based on results approaching statistical significance: “There were some specific HRV-PRO relationships that approached statistical significance and warrant further examination in larger prospective studies. For example, SDNN and depression in female patients, as well as those patients in the older age group (20-25y).” [Page 13]

Comment: Page 5 & 6 Table 1 & Line 12 My understanding is that ‘normal or reference ranges’ are usually indicated by mean � 2SD which encompasses most of the data, and not the ICR which only covers the middle 50% of the possible data values.

Change: Yes, we agree this is typically true. However, in the cited study (Sharma, et al) HRV for healthy adolescents was not normally distributed, and thus the data was reported as medians (and 25th – 75th percentiles). [Pages 2 and 10]

Comment: Page 7, Line 10 Suggest replace ‘7’ by ‘7.0’

Change: This has been changed.

Reviewer 2 Comments:

Comment: I would suggest to entirely remove Figure 1 from the manuscript. Instead, the authors should elaborate on the supposed pathways in the according paragraph of the introduction section, while citing relevant literature.

Change: We have removed Figure 1 and included additional text and references for relevant mechanistic pathways in biobehavioral oncology: “Psychosocial and environmental stressors can activate a cascade of signaling pathways that have overlapping behavioral and biologic implications. For example, a perceived threat (such as a new cancer diagnosis) triggers the hypothalamic-pituitary-adrenal (HPA) axis to ultimately secrete glucocorticoids into circulation, which have well-documented metabolic and immunomodulatory effects that influence cancer-related outcomes. The autonomic nervous system (ANS) is another key physiologic mediator of the relationship between patient experience and outcomes in biobehavioral oncology. Activation of the parasympathetic and sympathetic branches of the ANS has been associated with symptoms of anxiety, depression, and fatigue in patients with cancer. Conversely, interruption of sympathetic adrenergic signaling using β-blocker medications is associated with improved metastatic and inflammatory biomarkers, as well as lower cancer-specific mortality in patients with breast cancer.” [Page 3]

Comment: I was wondering whether the authors checked for differences in demographics / psychosocial outcomes between patients with and without available ECG recordings?

Change: We did not specifically look at differences by demographics or psychosocial outcomes, as in nearly half of the missing cases, pre-treatment ECGs were not medically necessary. This would leave a very small group (n=9 participants) to analyze, which would not likely yield meaningful results. We have now included documented reasons for missing ECGs: “Of the n=16 participants with missing ECGs, 7 did not require ECGs for their specific treatment plan, and there was no documented reason for omission in 9 cases.” [Page 5] 

Comment: Reporting HRV measurement, the authors may consider following and citing current guidelines: Quintana, D., Alvares, G. & Heathers, J. Guidelines for Reporting Articles on Psychiatry and Heart rate variability (GRAPH): recommendations to advance research communication. Transl Psychiatry 6, e803 (2016). https://doi.org/10.1038/tp.2016.73

Change: Thank you for this excellent reference. In addition to including the citation, we have updated our Methods section to more closely align with GRAPH standards: “HRV was derived using 10 second 100Hz ECGs obtained at the time of cancer diagnosis or relapse as part of routine clinical care. These screening ECGs were obtained as part of patients’ medical care and were not part of the clinical trial. Individual paper ECGs were retrospectively extracted from the medical record and scanned into a digital format using a high-resolution scanner. These digital ECG files were then uploaded into an image processing software (WebPlotDigitizer) to extract R-R intervals from the ECG tracing images. R-waves were first identified using the software algorithm and then manually reviewed for accuracy and artifacts. The discrete R-R intervals (in milliseconds) could then be identified from the image and converted to their numerical form. Using the open-source R software, RHRV, we then derived the two most widely used time domain parameters: standard deviation of normal to normal beats (SDNN) and root mean square of successive differences (RMSSD) per published guidelines.” [Pages 5-6]

Comment: Table 1 seems redundant, and I would suggest to remove it from the manuscript. In addition, Table 4 could be integrated in the supplementary table. Figure 2 also seems largely redundant, instead, the additional information provided could be integrated in Table 3. Comparisons with population norms might be depicted instead.

Change: We have now removed Table 1 and integrated the additional information from Figure 2 into Table 3 (now Table 2). We have also moved the histograms depicting HRV values compared to population norms from the Supplemental data to the main manuscript (now labeled Figure 1). As Table 4 (now Table 3) presents the primary data analysis for this study, we feel it is important to include this data in the main manuscript but have adjusted the reporting of the data to make the interpretation of ß coefficients more meaningful. To further ease reader interpretation, we have created scatter plots instead for easier visualization (now Figure 2).

Comment: Can the authors please clarify whether ECG recordings were collected before, at the same time as, or after psychosocial measures, and how much time elapsed between measurements?

Change: As this study was a secondary analysis, we did not have control over timing of ECG collection, and this varied in relation to survey administration. We have now incorporated this data into Table 1 and included the following statement in the Results section: “On average, surveys were collected 7 days after ECGs (rage 85 days before to 64 days after).” [Page 8]

Comment: In their discussion, the authors state "Importantly, as a secondary analysis of a larger trial, our study was not powered to detect our associations of interest". I was wondering whether the authors could explicate how the respective associations of interest should be investigated in future studies?

Change: We have now given more explicit suggestions for future study design: “Future investigations should prospectively collect extended (24h) HRV recordings in parallel with psychosocial patient reported outcome measures. The optimal study protocol would also avoid data collection during acute medical situations that could confound HRV results in patients with cancer (e.g. serious infection, chemotherapy toxicity, significant cardiac medication adjustments).” [Page 13]

Comment: In the discussion the authors also state "However, newly diagnosed patients also had median HRV parameters that appeared lower than healthy adolescents, raising the question of other non-treatment-related sources for autonomic dysregulation." I was wondering whether the authors might enrich their discussion by elaborating on potential "non-treatment-related sources for autonomic dysregulation" in this population, highlighting important research gaps.

Change: We have included specific examples of theorized non-treatment-related etiologies for autonomic dysregulation in AYAs with cancer: “However, newly diagnosed patients also had median HRV parameters that appeared lower than healthy adolescents, raising the question of other non-treatment-related sources for autonomic dysfunction that may come with a new cancer diagnosis. These could include disruption of normal neuro-cardiovascular physiology by the cancer itself (e.g. a large mass in the mediastinum or catecholamine-secreting tumor) or the intensified psychosocial stressor of the diagnosis in the AYA population (and accompanying social, developmental, financial, and existential threats). As HRV is easily measured through non-invasive methods, it could provide an innovative tool to help risk stratify patients who may require additional forms of supportive care.” [Page 12-13]

Comment: The R-package used (RHRV) should be properly cited to acknowledge the authors’ work:

Leandro Rodriguez-Linares, Xose Vila, Maria Jose Lado, Arturo Mendez, Abraham Otero and Constantino Antonio Garcia (2019). RHRV: Heart Rate Variability Analysis of ECG Data. R package version 4.2.5. https://CRAN.R-project.org/package=RHRV

Change: We have now included this citation. [Page 6]

Comment: For group comparisons, besides p-values the authors may also report according effects sizes.

Change: We do agree that effect sizes generally assist with interpretability; however, in the case of the nonparametric testing that is required of comparing HRV population values to our study cohort, the effect sizes do not necessarily provide meaningful information. And so although we can provide numerical effect sizes from the modified ranksum testing performed, we feel that in this case comparing raw differences in medians (and IQRs) gives readers a clearer sense of how the two groups might be different.

---

## [Decision Letter · Decision Letter 1]

19 Oct 2021

Heart Rate Variability and Psychosocial Symptoms in Adolescents and Young Adults with Cancer

PONE-D-21-16445R1

Dear Dr. Taylor,

We’re pleased to inform you that your manuscript has been judged scientifically suitable for publication and will be formally accepted for publication once it meets all outstanding technical requirements.

Kind regards,

Michael Kaess, M. D.

Academic Editor

PLOS ONE

Additional Editor Comments (optional):

Reviewers' comments:

Reviewer's Responses to Questions

**Comments to the Author**

1. If the authors have adequately addressed your comments raised in a previous round of review and you feel that this manuscript is now acceptable for publication, you may indicate that here to bypass the “Comments to the Author” section, enter your conflict of interest statement in the “Confidential to Editor” section, and submit your "Accept" recommendation.

Reviewer #1: All comments have been addressed

Reviewer #2: All comments have been addressed

2. Is the manuscript technically sound, and do the data support the conclusions?

Reviewer #1: Yes

Reviewer #2: Yes

3. Has the statistical analysis been performed appropriately and rigorously? 

Reviewer #1: Yes

Reviewer #2: Yes

4. Have the authors made all data underlying the findings in their manuscript fully available?

Reviewer #1: Yes

Reviewer #2: Yes

5. Is the manuscript presented in an intelligible fashion and written in standard English?

Reviewer #1: Yes

Reviewer #2: Yes

6. Review Comments to the Author

Reviewer #1: The addition of Figure 2 adds substantially to the previous version.

All my comments can be released to the authors

Reviewer #2: I would like to thank the authors for their careful revision and for addressing all my comments. Elaborations on why certain suggestions were not implemented seem plausible to me, and I think the manuscript has improved much during the revision process. In addition, it is great that the underlying dataset has now been shared by the authors! All in all, I think this study is now suitable for publication.

7. PLOS authors have the option to publish the peer review history of their article (what does this mean?). If published, this will include your full peer review and any attached files.

Reviewer #1: No

Reviewer #2: No

---

## [Editor Report · Acceptance letter]

25 Oct 2021

PONE-D-21-16445R1 

Heart rate variability and psychosocial symptoms in adolescents and young adults with cancer 

Dear Dr. Taylor:

I'm pleased to inform you that your manuscript has been deemed suitable for publication in PLOS ONE. Congratulations! Your manuscript is now with our production department. 

Kind regards, 

on behalf of

Prof. Dr. Michael Kaess 

Academic Editor

PLOS ONE